# Analysis of Nursing Students’ Nonverbal Communication Patterns during Simulation Practice: A Pilot Study

**DOI:** 10.3390/healthcare11162335

**Published:** 2023-08-18

**Authors:** Eunju Jin, Hyunju Kang, Kunyoung Lee, Seung Gun Lee, Eui Chul Lee

**Affiliations:** 1Department of Nursing, Gangneung Yeongdong University, Gangneung-si 25521, Republic of Korea; 201915076@kangwon.ac.kr; 2College of Nursing, Kangwon National University, Chuncheon-si 24341, Republic of Korea; 3Department of Computer Science, Graduate School, Sangmyung University, Jongno-gu, Seoul 03016, Republic of Korea; 201933048@sangmyung.kr; 4Department of AI & Informatics, Graduate School, Sangmyung University, Jongno-gu, Seoul 03016, Republic of Korea; 202233053@sangmyung.kr; 5Department of Human-Centered Artificial Intelligence, Sangmyung University, Jongno-gu, Seoul 03016, Republic of Korea; eclee@smu.ac.kr

**Keywords:** nursing students, nonverbal communication, facial expression, simulation training, pilot study, nursing education

## Abstract

Therapeutic communication, of which nonverbal communication is a vital component, is an essential skill for professional nurses. The aim of this study is to assess the possibility of incorporating computer analysis programs into nursing education programs to improve the nonverbal communication skills of those preparing to become professional nurses. In this pilot observational study, the research team developed a computer program for nonverbal communication analysis including facial expressions and poses. The video clip data captured during nursing simulation practice by 10 3rd- and 4th-grade nursing students at a university in South Korea involved two scenarios of communication with a child’s mother regarding the child’s pre- and post-catheterization care. The dominant facial expressions varied, with sadness (30.73%), surprise (30.14%), and fear (24.11%) being the most prevalent, while happiness (7.96%) and disgust (6.79%) were less common. The participants generally made eye contact with the mother, but there were no instances of light touch by hand and the physical distance for nonverbal communication situations was outside the typical range. These results confirm the potential use of facial expression and pose analysis programs for communication education in nursing practice.

## 1. Introduction

Therapeutic communication (TC), a fundamental aspect of the nurse–patient relationship, involves understanding a patient’s situation and the need to provide appropriate care [1,2]. When nurses use effective TC skills, they not only foster therapeutic relationships with patients and enhance patient satisfaction but also improve the overall quality of care [2,3,4]. In addition, patients and their families consider it important for nurses to listen attentively, engage in meaningful conversations, and communicate information in an empathetic manner through TC [2,3].

TC has both verbal and nonverbal aspects [4]. Nonverbal communication is a vital component of TC and includes all forms of meaningful communication that cannot be conveyed through verbal language. Only 7% of message delivery is attributable to verbal messages, with nonverbal messages predominating [5]. Nonverbal communication encompasses a range of important factors including facial expressions, eye contact, touch, body language, gestures, pose, and tone of voice [3,6,7]. Facial expressions are often associated with specific emotions, making them an important aspect of nonverbal communication [8]. Humans experience a range of emotions that are often accompanied by specific facial expressions. These include, but are not limited to, anger, happiness, sadness, surprise, fear, and disgust [9,10]. Various types of touch and poses, such as placing a hand lightly on a patient’s shoulder, holding their hand, or coming physically closer, might be used as a means of showing empathy and reassuring patients [11].

Effective TC is a crucial skill for nursing students preparing to become professional nurses, and it is important that they receive adequate training and education to develop their professional communication competencies. However, nursing students often face challenges and stress when communicating with patients and their families [12]. In particular, nursing students may have inadequate experience with face-to-face communication because of their reliance on online communication and messaging. Therefore, in order to improve nursing students’ communication skills, nursing education programs utilize various simulation practices, such as roleplay and standardized patients [9,13]. These educational interventions have been reported to lead to improvements in the communication competencies of nursing students [13]. However, most training and education programs lack the component of nonverbal communication skills [14].

When evaluating nursing students’ communication skills, instructors usually assess their level of confidence in communication and their verbal communication abilities through direct observation [13]. Although some nonverbal communication items are included in the checklist for evaluating communication skills, it is challenging to assess specific aspects such as facial expressions and poses. In particular, it is difficult for students and instructors to objectively view and evaluate, respectively, nonverbal communication, which is an important aspect of the development of communication skills. Therefore, there are limitations to incorporating and utilizing nonverbal communication in nursing education.

Recently, a computer program that utilizes deep learning to recognize personal facial expressions has been developed and commercialized [15]. Another study evaluated communication effectiveness during on-camera media interview training using automatic nonverbal recognition systems [16]. These programs have been adopted in some nursing studies to assess the distress of patients with cancer and explore the potential correlation between test anxiety and the facial expressions of nursing students [8,17]. Another study has investigated variations in facial expressions among nursing students in response to various scenario-based situations [9]. However, we could not identify any studies that have used computer analysis programs to validate nursing students’ nonverbal communication with a specific focus on poses and facial expressions.

Against this background, this study aimed to assess the possibility of incorporating computer analysis programs into nursing practice education to improve students’ nonverbal communication skills. By conducting a pilot study that objectively evaluated the nonverbal communication patterns of nursing students, including their facial expressions and poses, we aimed to explore the potential of using these programs in future nursing education.

## 2. Materials and Methods

### 2.1. Study Design and Sample

In this pilot observational study, we conducted a secondary computer analysis of a video clip taken during simulation nursing practice education.

The sample for this study consisted of video clip data captured during the nursing simulation practice of 10 3rd- and 4th-grade nursing students at a university in South Korea. The topic of the simulation scenario was related to nursing care for a child and mother after the child had undergone pre- and post-cardiac catheterization. Cardiac catheterization is the insertion and passage of a small plastic tube into the arteries and veins to perform coronary angiography or catheter-based treatments of structural heart diseases like arterial septal defect [18]. The participating nursing students acted as nurses and had to communicate with the standardized patient, who played the role of the child’s mother.

### 2.2. Facial Expression and Pose Analysis Program

First, our research team developed a computer program to analyze the students’ facial expressions and body language as a pose. Our face analysis program uses machine learning algorithms based on face detection, enabling the accurate detection and analysis of the participants’ facial expressions. Based on the analytical model of facial affect [19], the program was able to extract seven basic emotional states, namely, neutral, happy, sad, surprised, fearful, disgusted, and angry. The program assigned a value between 0 and 1 to each of these seven basic emotions, with a higher value indicating a greater likelihood of a facial expression representing the emotion. The average value of the statistics for each frame was calculated, and statistics with a confidence value of 0.8 or higher were selected for analysis. The confidence level was calculated by considering camera and face angle because facial recognition can produce inaccurate results if the camera does not capture a face from the front.

Second, we developed a pose analysis program based on four key nonverbal communication skills in nursing care [3,6]. The first pose skill that we analyzed concerned whether the nurse made eye contact with the mother during the conversation. We analyzed the angles of both the nurse and mother’s gazes, with a smaller angle indicating that the nurse and the mother were closer to making eye contact. The second skill concerned the use of light touch. We measured the distance between the nurse’s hand and the mother, with a smaller distance ratio indicating that the nurse’s hand was closer to the mother, and therefore closer to a light touch. The third skill concerned whether an appropriate distance was maintained during the conversation. We measured the physical distance between the nurse and the mother in centimeters. Finally, we analyzed whether the nurse’s upper body tilted toward the mother during the conversation. We measured the angle between the direction perpendicular to the ground and the nurse’s waist, with 180° indicating that the nurse did not bend and a smaller angle indicating that the nurse’s waist was more inclined. Table 1 and Figure 1 list the range of analytical methods and the values for each pose analysis item.

### 2.3. Simulation Setting and Data Collection

The nursing simulation practice focused on providing nursing care for a five-year-old child who had been diagnosed with atrial septal defects and admitted to the pediatric ward for cardiac catheterization. The simulation practice consisted of two scenarios entailing communication with the child’s mother regarding pre- and post-catheterization care.

In the first simulation scenario, regarding pre-care for catheterization, the nursing students had to communicate with the mother, who was anxious about her child’s delayed procedures. Students were required to answer questions and address the mother’s concerns effectively. In the second simulation scenario, regarding post-care, the focus was on nursing practice that educates and supports the mother after catheterization. The students had to communicate appropriately in order to provide the necessary information and support to the mother.

The mother was a professional actor acting as a standardized patient and was trained to ask the same questions and maintain consistency with all the students. The students were expected to communicate with her in the role of a nurse. They were educated about this scenario but were not given specific guidance on how to communicate verbally and nonverbally in these scenarios.

A camera was installed behind the mother to record video clips of the communications. The recording lasted for approximately five minutes, starting from the beginning of the communication. We analyzed the video clips of all 10 nursing students from their first scenario-based simulation practice to perform facial expression analysis and used video clips from their second simulation practice to conduct pose analysis.

### 2.4. Data Analysis

Descriptive statistics were calculated using SPSS statistical software (version 25.0; IBM Corp., Armonk, NY, USA). We analyzed each of the 10 video clips, recorded before and after pediatric cardiac catheterization, to assess facial expression and poses. Facial expressions were analyzed once every 30 frames. To ensure the reliability of the facial expression analysis data, we calculated these data based on the camera and face angle, which had a range of 0–1. The closer the reliability value was to 1, the higher the likelihood that the front of the face would be accurately captured by the camera, indicating data with high reliability. We only used statistics from facial expression analysis data with a confidence level of 80% or higher. The statistics of the seven facial expressions were calculated as values ranging from 0 to 1. The closer a value was to 1, the more likely it was that the expression conveyed the relevant emotion. We obtained the average value of the 10 participants’ facial expressions and expressed that value as a percentage. The percentage of facial expressions for the corresponding emotions was analyzed from the video clips. For the pose analysis, we obtained the average value of the four pose items in each video clip, and then calculated the average value for all the video clips.

### 2.5. Ethical Considerations

This study is a secondary analysis of video clips taken from simulation practice education programs. We conducted this study after obtaining consent from the students to include them in research and video recording; therefore, we obtained a review exemption permit (IRB-2023-04-008).

## 3. Results

Table 2 shows the general characteristics of the participating nursing students. We included nursing students aged 20–23 years, most of whom were women. Approximately 50% of the students reported being satisfied with their majors, while 20% said that they had good interpersonal relationships. None of the participants had received formal communication-related education.

Table 3 shows the results of the facial expression analysis program. The distribution of the seven facial expressions varied across participants, with participants No. 3 and 4 showing the highest levels of surprise and No. 5 and 9 showing the highest levels of fear. Few participants displayed expressions of anger during the study.

Figure 2 shows the average proportion of facial expressions for each participant during a nursing simulation practice involving a conversation with a mother whose child was about to undergo cardiac catheterization. The most frequently observed facial expressions involved sadness (30.73%), surprise (30.14%), fear (24.11%), and neutral feelings (13.29%). Happiness (7.96%) and disgust (6.79%) were less common, while anger (0.05%) was rare.

Table 4 presents the results of the pose analysis program. The analysis measured four items. First, the average angle of eye contact between the participants and the mother was 64.19 ± 20.35°. Second, the average distance ratio between the participants’ hands and the mother was 0.84 ± 0.03. Third, the average distance between the participants and the mother was 159.72 ± 27.22 cm, with some participants moving closer and others moving further away. Finally, the average upper-body bending angle was 151.24 ± 6.37°, with a smaller value indicating a more inclined waist.

## 4. Discussion

The ability to use nonverbal communication effectively is an important skill for nurses as it can enhance trust in nurse–patient relationships, promote patients’ participation in their own care, and improve health outcomes [20]. A systematic review and meta-analysis conducted to examine the effects of nonverbal communication among medical staff on patient outcomes in communication situations found that patients were more satisfied when the medical staff listened to them carefully [3]. Accordingly, educational interventions have been performed to improve the communication skills of nursing students, and significant results have been obtained [4]. However, it is difficult to find studies that have proposed educational strategies to improve nonverbal communication skills, which are important in therapeutic communication. In this study, before providing educational interventions to improve nursing students’ nonverbal communication skills, we explored how to objectively evaluate and analyze nonverbal communication skills and the possibility of their use.

In this pilot study, we explored the potential use of nonverbal communication in nursing education to improve students’ skills. Specifically, we analyzed nursing students’ nonverbal communication during child nursing simulation practice as they communicated with a mother. The nursing students engaged in a five-minute conversation with a mother who was anxious about her child’s upcoming cardiac catheterization. Facial expressions were recorded and analyzed once every 30 frames in a video clip, and only results with a reliability of 0.8 or higher were included in the analysis. The distribution of the seven facial expressions was examined, with the participants found to exhibit a variety of expressions. The dominant facial expressions varied among the participants; however, on average, sadness (30.77%), surprise (30.15%), and fear (29.01%) were most prevalent. In contrast, happiness (7.88%) and disgust (6.83%) were less common. Overall, there was a wide range of generally negative facial expressions. The scenario involved the worried mother of a child who expressed concern about the delayed catheterization schedule. The participants’ reactions were likely influenced by their understanding and empathy toward the mother’s situation. Neutral expressions accounted for 13.29% of all the facial expressions observed. To understand the underlying meaning of this expression, it would be helpful to examine the specific moments in the conversation between the participant and the mother when the neutral expression was displayed. Our results indicated that anger, as a negative emotion, was rarely expressed during simulation practice. Overall, the nursing students exhibited a range of facial expressions during the communication scenario, and the objective measurements used in this study can be used for self-reflection and feedback for students in future debriefings and education classes.

The next step in our study involved filming the communication between the nursing student and the mother for approximately five minutes. The scenario involved the return of the child’s mother to the hospital room after cardiac catheterization and her expressing worry that the child had not yet woken up. Four behaviors commonly recommended for nonverbal communication skills training were analyzed using the computer program [6]. These included maintaining eye contact during face-to-face conversations with a patient, using light touch, maintaining an appropriate distance during conversations, and tilting the upper body toward the patient during conversations. In our study, the participants’ eye contact angle was found to be 64.19 ± 20.35° (with a reference value of 0–180°), and they generally made eye contact with the mother. However, the frequency of light touch was only 0.84 ± 0.03 (with a reference value ratio of 0–1), and there were no instances of light touch by hand during the filming period. The appropriate physical distance for nonverbal communication situations is typically 45–120 cm (with reference). In this study, the average physical distance between the participants and the mother was 159.72 ± 27.22 cm, which lies outside the typical range. However, it should be noted that during the filming period, there were instances where the nursing student left to pick up an item or crossed in front of the mother to turn off the monitor. Therefore, when analyzing a recorded pose, it is important to consider situations in which maintaining a consistent communication distance may be difficult, such as when deviating from the camera angle during a conversation. Finally, we also measured the degree of bending toward the mother in the participants’ upper body during communication. The results showed that, on average, the nursing students slightly or barely bent their upper body toward the mother (151.24 ± 6.37°, with a reference value 0–180°). Bending the body forward toward the other person shows that the listener is actively engaged in listening to the speaker [6,7].

In a study that compared the nonverbal communication skills of nursing students and nurses, it was found 90% of the nursing students tended to stand during communication [21]. Furthermore, the study revealed that nursing students made less eye contact with patients than did nurses, and that their upper limb or hand movements were shorter in duration. These results are consistent with our findings. It would be useful to utilize these findings to provide feedback on listening pose and incorporate analysis of upper-body bending and the degree of eye contact into nonverbal communication training programs for nursing students. In another study assessing the nonverbal communication skills of internationally educated nurses in the United States, a checklist of 12 nonverbal communication behaviors was developed and used to measure their interactions with standardized patients [22]. Low scores were obtained for behaviors such as leaning forward, lowering the body, hugging, shaking hands, and providing a therapeutic touch. The highest scores were obtained for nonverbal communication behaviors such as not making distracting movements, maintaining eye contact, and smiling. These observer-evaluated results are comparable to those obtained using a computer program in our study. The results of computer-based facial and pose analysis could help to evaluate the nonverbal communication patterns of nursing students and could be used for educational feedback. However, rather than approaching it as an absolute standard, it should be noted that nonverbal communication can be understood differently depending on the cultural context [23].

A growing number of studies is focusing on deep learning-based face recognition technology, which has significantly improved accuracy compared with previous methods [15]. A nursing study reported the potential use of facial expression recognition programs as complementary tools to enhance the accuracy of distress assessments in patients with cancer [17]. Another study confirmed the relationship between nursing skills, test anxiety, self-efficacy, and facial expressions among nursing students [8]. However, we were unable to identify any research that examines nonverbal communication exhibited by nursing students in simulated patient interactions and explores its potential application in nursing education. In this pilot study, we aimed to evaluate the nonverbal communication patterns of nursing students in a simulation practice using deep learning-based facial expression recognition and pose analysis. The findings suggest that such tools could be utilized to enhance nonverbal communication skills in nursing education in the future. As this is a pilot study of only 10 samples, there are some suggestions for the direction of future research. First, expanding the sample size would provide researchers with more representative data. Second, when using this facial and pose analysis program, including different scenarios or patient populations would offer a broader perspective on how nonverbal communication skills can be improved. Third, researchers should consider including additional aspects of nonverbal communication, such as tone of voice or body language, to gain a more comprehensive understanding of nursing students’ skills and potential areas for improvement.

The strength of this study is its use of deep learning-based technology to analyze nonverbal communication patterns, including facial expressions and poses. This approach offers promising possibilities for future nursing education programs. However, this study has limitations in terms of generalizing its results owing to the small number of participants. To increase its external validity, future studies should include larger sample sizes. Furthermore, because the analysis was only computer-based and performed without other comparisons, the reliability of the results was limited. Therefore, we suggest conducting a future study that compares and analyzes recorded data using both computer-based and human observer evaluations. Additionally, because of participant movement during the filming, it was difficult to fully comprehend the consistent nonverbal communication patterns in this study. Thus, we propose the development and application of nursing care scenarios that allow for the practice and refinement of nonverbal communication skills.

## 5. Conclusions

After analyzing nonverbal communication patterns using a deep learning-based facial expression and pose analysis program during nursing simulation practice, we found that nursing students exhibited various facial expressions when communicating with a distressed mother, with anger rarely exhibited. Although eye contact was generally maintained, the results revealed the need to address other aspects of nonverbal communication such as maintaining an appropriate distance, using appropriate touch, and tilting the upper body toward the patient. Overall, this pilot study confirmed the potential use of facial expression and pose analysis programs for communication education in nursing practice.

## Figures and Tables

**Figure 1 healthcare-11-02335-f001:**
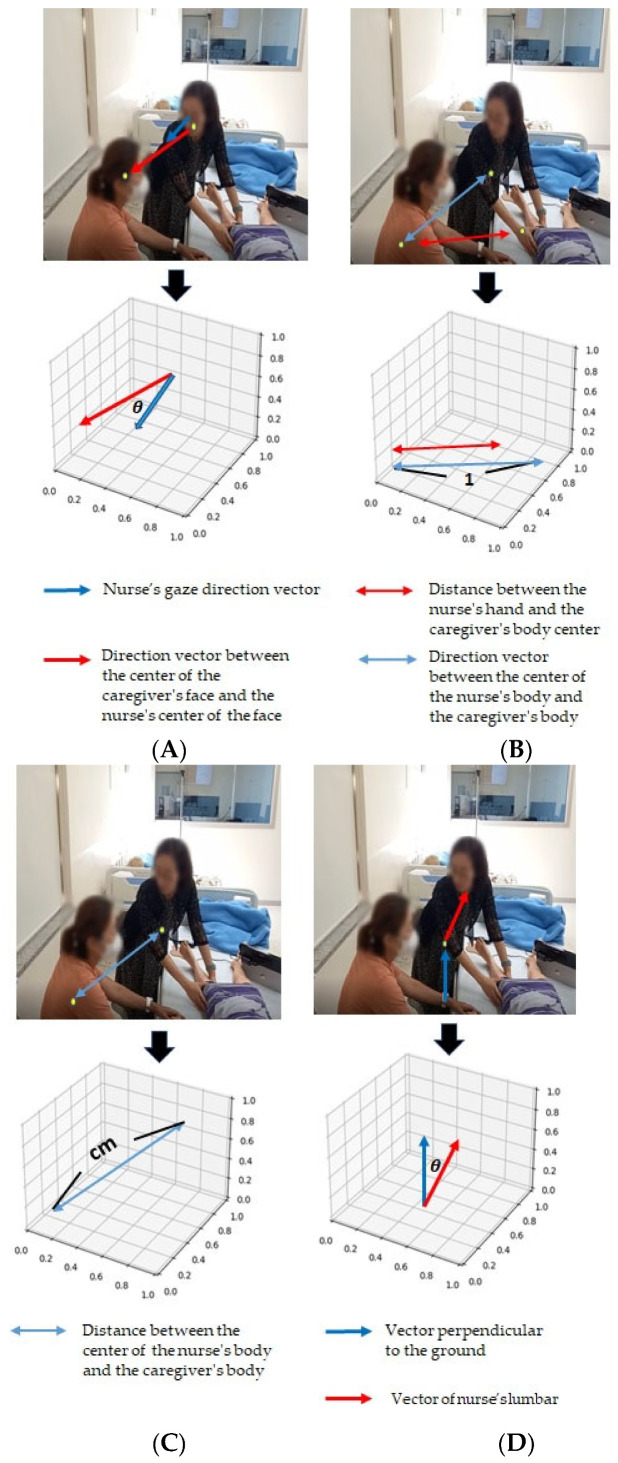
Examples of the pose analysis program. (**A**) Example of pose analysis item 1 (Eye-contact); (**B**) Example of pose analysis item 2 (Light touch); (**C**) Example of pose analysis item 3 (Distance); (**D**) Example of pose analysis item 4 (Titled upper body).

**Figure 2 healthcare-11-02335-f002:**
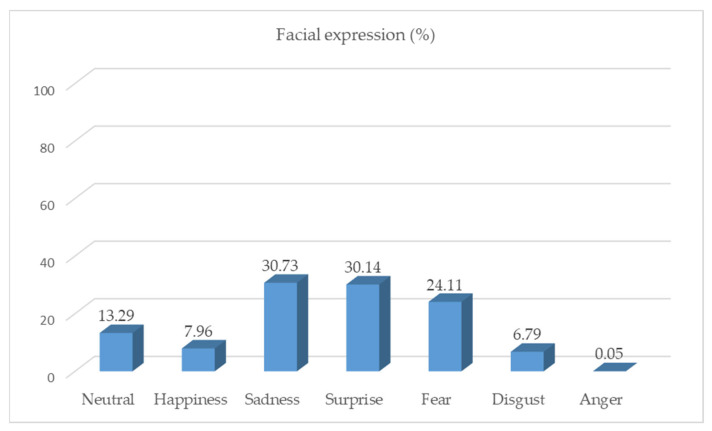
The overall distribution of facial expressions.

**Table 1 healthcare-11-02335-t001:** Items in the pose analysis program.

Item	Description of Analysis	Range
1. Eye contact	Angles of both the nurse and mother’s gaze	0–180°
2. Light touch	Distance between the nurse’s hand and the mother (ratio of distance values between the nurse and the mother)	0–1
3. Distance	Physical distance between the nurse and the mother	Centimeter
4. Tilted upper body	Angle between the direction perpendicular to the ground and the nurse’s waist	0–180°

**Table 2 healthcare-11-02335-t002:** General characteristics of the participants (N = 10).

Participant	Age (Years)	Gender	Satisfaction with Major	Interpersonal Relations	Communication Education Experience
1	21	Woman	Satisfied	Moderate	No
2	21	Man	Satisfied	Good	No
3	23	Woman	Moderate	Good	No
4	20	Woman	Satisfied	Moderate	No
5	20	Woman	Moderate	Moderate	No
6	20	Woman	Moderate	Moderate	No
7	22	Woman	Satisfied	Moderate	No
8	20	Woman	Moderate	Moderate	No
9	21	Woman	Moderate	Moderate	No
10	20	Woman	Satisfied	Moderate	No

**Table 3 healthcare-11-02335-t003:** Results of the facial expression program (N = 10).

Participant	Neutral	Happiness	Sadness	Surprise	Fear	Disgust	Anger
	(%)	(%)	(%)	(%)	(%)	(%)	(%)
1	5.40	23.40	51.00	0.00	0.10	11.40	0.00
2	0.10	0.80	0.30	0.10	49.00	0.40	0.00
3	2.80	1.30	15.40	180.10	0.40	8.00	0.10
4	10.60	0.60	79.80	117.60	3.90	3.10	0.00
5	2.10	0.30	21.40	0.30	75.30	0.10	0.10
6	33.30	34.70	24.20	0.00	7.00	0.20	0.00
7	40.20	1.80	20.50	0.50	33.20	0.00	0.00
8	18.60	5.80	57.50	0.00	5.40	4.40	0.10
9	12.10	10.70	11.50	2.80	59.40	0.10	0.20
10	7.70	0.20	25.70	0.00	7.40	40.20	0.00

**Table 4 healthcare-11-02335-t004:** Results of the pose expression program (N = 10).

Participant	Eye Contact	Light Touch	Distance	Tilted Upper Body
	Mean ± SD (Min–Max)	Mean ± SD (Min–Max)	Mean ± SD (Max–Min)	Mean ± SD (Max–Min)
1	52.72 ± 11.91 (19.31–78.09)	0.78 ± 0.08 (0.56–0.97)	135.25 ± 13.07 (104.25–171.94)	147.51 ± 7.42 (131.96–171.03)
2	59.2 ± 31.74 (12.27–151.33)	0.87 ± 0.11 (0.56–1.31)	125.04 ± 42.64 (31.27–281.66)	151.12 ± 9.3 (127.59–175.44)
3	54.84 ± 17.95 (8.31–157.93)	0.83 ± 0.08 (0.63–1.15)	143.11 ± 54.74 (58.99–402.34)	160.14 ± 6.93 (132.67–177.7)
4	62.5 ± 28.82 (13.52–160.53)	0.88 ± 0.1 (0.7–1.33)	231.15 ± 107.35 (20.01––388.06)	149.05 ± 16.54 (107.38–173.2)
5	71.77 ± 27.95 (6.72–151.62)	0.86 ± 0.23 (0.56–2.26)	237.54 ± 66.94 (−10.15–374.77)	136.9 ± 20.35 (108.62–173.08)
6	45.89 ± 20.24 (4.93–147.18)	0.78 ± 0.13 (0.52–1.04)	196.84 ± 65.76 (87.48–457.78)	148.57 ± 10.07 (129.98–170.73)
7	73.08 ± 35.27 (14.57–165.5)	0.86 ± 0.11 (0.53–1.47)	154.4 ± 89.34 (−8.95–355.49)	154.72 ± 14.45 (109.82–178.91)
8	74.8 ± 32.51 (20.29–146.35)	0.88 ± 0.14 (0.67–1.3)	95.28 ± 59.04 (−1.73–208.29)	153.19 ± 14.74 (128.69–178.82)
9	84.68 ± 34.99 (19.11–156.18)	0.85 ± 0.15 (0.48–1.44)	155.73 ± 81.06 (6.93–324.43)	155.83 ± 9.78 (134.16–178.63)
10	62.47 ± 38.09 (15.55–156.01)	0.8 ± 0.11 (0.63–1.2)	122.84 ± 36.87 (45.48–261.93)	155.41 ± 10.64 (135.69–175.75)
Total	64.19 ± 20.35 (4.93–165.5)	0.84 ± 0.03 (0.48–2.26)	159.72 ± 27.22 (−10.15–457.78)	151.24 ± 6.37 (107.38–178.91)

## Data Availability

The data that support the findings of this study are available on request from the corresponding author.

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
