# Peer review of "Analysis of Nursing Students’ Nonverbal Communication Patterns during Simulation Practice: A Pilot Study"

_healthcare, 2023, doi:10.3390/healthcare11162335_

Round 1

Reviewer 1 Report

This study aims to explore the possibility of incorporating computer analysis programs into nursing education programs to enhance the nonverbal communication skills of future professional nurses. The researchers conducted a pilot observational study where they developed a computer program for analyzing nonverbal communication, specifically focusing on facial expressions and poses.

I would like to express my appreciation for your informative research work and commend you on the presentation of your findings. I found it to be extremely informative and well-executed. However, I have a few minor observations and comments that may help further enhance your research work. Please consider addressing the following points in detail

Based on your feedback, here are the revised comments for the authors:

Dear Authors,

I would like to express my appreciation for your informative research work and commend you on the presentation of your findings. I found it to be extremely informative and well-executed. However, I have a few minor observations and comments that may help further enhance your research work. Please consider addressing the following points in detail:

The abstract should not only describe the paper's accomplishments but also highlight the significance of the results obtained.

In the introduction section, it would be beneficial to prominently outline the specific contributions of your work using bullet points or a concise summary.

 It is recommended to include a dedicated "Literature Review" section or subsection in your paper. This section should provide a comprehensive comparison of existing models through tables or other means to identify their strengths and weaknesses accurately. Additionally, please ensure that recent studies from 2020 - 2023 are included in this literature review, particularly regarding formulation and pollution aspects.

To strengthen your study's credibility, consider comparing your results with relevant existing studies. These references will enable readers to appreciate how your findings contribute uniquely to this field.

Within the methodology section, consider including a subsection explicitly defining all the medical definitions used throughout the paper for better clarity and understanding by readers.

 It is essential to improve figure quality throughout the article as they play a crucial role in conveying information effectively.

 I recommend revising the formatting of your paper thoroughly to ensure consistency and adherence to appropriate guidelines.

Something recommendations for author to consider.

1. Increase sample size: Since this was a pilot study involving only 10 nursing students from one university in South Korea, expanding the sample size would provide more representative data.

2. Diversify scenarios: Including different scenarios or patient populations would offer a broader perspective on how nonverbal communication skills can be improved.

3. Incorporate qualitative analysis: While this study focused primarily on quantitative analysis of facial expressions and poses, integrating qualitative methods such as interviews or surveys could provide deeper insights into student experiences related to nonverbal communication.

4.Expand assessment criteria: Consider including additional aspects of nonverbal communication such as tone of voice or body language to gain a more comprehensive understanding of the participants' skills and potential areas for improvement.

5. Longitudinal study design: Conducting a longitudinal study that follows nursing students over time could provide insights into the effectiveness of incorporating computer analysis programs into nursing education and its impact on their long-term nonverbal communication skills.

Careful review is required. 

Author Response

We thank you and the reviewers for your thoughtful suggestions and insights. 

Reviewer’s Comments and Suggestions Reviewer 1 :

 This study aims to explore the possibility of incorporating computer analysis programs into nursing education programs to enhance the nonverbal communication skills of future professional nurses. The researchers conducted a pilot observational study where they developed a computer program for analyzing nonverbal communication, specifically focusing on facial expressions and poses.

I would like to express my appreciation for your informative research work and commend you on the presentation of your findings. I found it to be extremely informative and well-executed. However, I have a few minor observations and comments that may help further enhance your research work. Please consider addressing the following points in detail

Based on your feedback, here are the revised comments for the authors:

Dear Authors,

I would like to express my appreciation for your informative research work and commend you on the presentation of your findings. I found it to be extremely informative and well-executed. However, I have a few minor observations and comments that may help further enhance your research work. Please consider addressing the following points in detail:

The abstract should not only describe the paper's accomplishments but also highlight the significance of the results obtained.

→(Line 28,29) We described the differentiated strengths of this study and the usefullness our analysis program.

In the introduction section, it would be beneficial to prominently outline the specific contributions of your work using bullet points or a concise summary.

→(Line 35-92) We revised the introduction section as follows;

  1. Importance of therapeutic communication (TC) skills in nursing
  2. Aspect of vebal and nonverbal communication in TC
  3. Necessity of nonverval communicaton education
  4. Recent research on facial expression or other body language using computer programs
  5. The purpose of this study (exploring the possibility of computer anlaysis of nonverbal communicatio in nursing practice education)

It is recommended to include a dedicated "Literature Review" section or subsection in your paper. This section should provide a comprehensive comparison of existing models through tables or other means to identify their strengths and weaknesses accurately. Additionally, please ensure that recent studies from 2020 - 2023 are included in this literature review, particularly regarding formulation and pollution aspects.

→(Line 52-54, 64-65,76-78) Due to the limitation of the number of letters, a literature review was included in the Introduction section and additionally reviewed and described three papers that published in 2020~2021.

To strengthen your study's credibility, consider comparing your results with relevant existing studies. These references will enable readers to appreciate how your findings contribute uniquely to this field.

→(Line 76-78, 292-296) We added the recent study that conducted to evaluate the communication effectiveness during on-camera media interview training using automatica nonverbal recognition systems [16].

Within the methodology section, consider including a subsection explicitly defining all the medical definitions used throughout the paper for better clarity and understanding by readers.

→(Line 100-103)The medical term in our manuscript is cardiac catheterization. We described the definition of cardiac catheterization in Material and Methods section.

It is essential to improve figure quality throughout the article as they play a crucial role in conveying information effectively.

à(Line 135-136) Figure 1 was revised to make it show more clear.

I recommend revising the formatting of your paper thoroughly to ensure consistency and adherence to appropriate guidelines.

→(Line 93-183 )Material and Methods section was revised as follws:

2.1. Study desgin and Sample, 2.2. Facial Expression and Pose Analysis Program, 2.3. Simulation Setting and Data Collection, 2.4 Data anlaysis, 2.5 Simulation Setting and Data Collection

Something recommendations for author to consider.

  1. Increase sample size: Since this was a pilot study involving only 10 nursing students from one university in South Korea, expanding the sample size would provide more representative data.
  2. Diversify scenarios: Including different scenarios or patient populations would offer a broader perspective on how nonverbal communication skills can be improved.
  3. Incorporate qualitative analysis: While this study focused primarily on quantitative analysis of facial expressions and poses, integrating qualitative methods such as interviews or surveys could provide deeper insights into student experiences related to nonverbal communication.
  4. Expand assessment criteria: Consider including additional aspects of nonverbal communication such as tone of voice or body language to gain a more comprehensive understanding of the participants' skills and potential areas for improvement.
  5. Longitudinal study design: Conducting a longitudinal study that follows nursing students over time could provide insights into the effectiveness of incorporating computer analysis programs into nursing education and its impact on their long-term nonverbal communication skills.

→(Line 309-316) I appreciate your suggestions. In Discussion section, I added this suggestions for future research.

Reviewer 2 Report

The article is interesting, but the study could be improve by increasing the number of study participants

Author Response

We thank you and the reviewers for your thoughtful suggestions and insights. 

Reviewer 2's coments : The article is interesting, but the study could be improve by increasing the number of study participants.

Author's replay: (Line 309-316) I appreciate your suggestions. In Discussion section, I added the limitation of our pilot study and the suggestions for future research to improve this study.

Reviewer 3 Report

Congratulations on this piece of research. The analysis of non-verbal communication using software is extremely original. I also agree about the benefits of training nurse students in non-verbal communication skills. The study you present is a pilot, so the results cannot be generalizabe. In fact, my impression is that there is not enough discussion about the particular aspects of non-verbal communication that you are studying (why those and no others, why is it important to classify facial expressions, etc.). As this is presented as a pilot, I would focus more on the methodology of the study and present a discussion that deals more with its appropriateness and possible applications to further studies. 

Author Response

We thank you and the reviewers for your thoughtful suggestions and insights. 

Reviewer’s Comments and Suggestions Reviewer 3 :

 Congratulations on this piece of research. The analysis of non-verbal communication using software is extremely original. I also agree about the benefits of training nurse students in non-verbal communication skills.

The study you present is a pilot, so the results cannot be generalizabe. In fact, my impression is that there is not enough discussion about the particular aspects of non-verbal communication that you are studying (why those and no others, why is it important to classify facial expressions, etc.). As this is presented as a pilot, I would focus more on the methodology of the study and present a discussion that deals more with its appropriateness and possible applications to further studies.

→(Line 309-316) I appreciate your suggestions. In Discussion section, I added the limitation of our pilot study and the suggestions for future research to improve this study.

→(Line 52-54) It was added that there were various type of nonverval communication for therapeutic communication.

→(Line 292-296) We described that the understanding of nonverbal communication may vary depending on cultural differences. 

Round 2

Reviewer 1 Report

Thank you for addressing my concern; I still believe there is room for improvement with the help of recent literature. 

Sound scientific.